# Enhanced Production and Functional Characterization of Recombinant Equine Chorionic Gonadotropin (rec-eCG) in CHO-DG44 Cells

**DOI:** 10.3390/biom15020289

**Published:** 2025-02-14

**Authors:** Munkhzaya Byambaragchaa, Sei Hyen Park, Myung-Hum Park, Myung-Hwa Kang, Kwan-Sik Min

**Affiliations:** 1Carbon-Neutral Resources Research Center, Hankyong National University, Anseong 17579, Republic of Korea; munkhzaya_b@yahoo.com; 2Institute of Genetic Engineering, Hankyong National University, Anseong 17579, Republic of Korea; 3Graduate School of Animal Biosciences, Hankyong National University, Anseong 17579, Republic of Korea; mrtree119@naver.com; 4TNT Research, Sejong 30141, Republic of Korea; pmh@tntresearch.co.kr; 5Department of Food Science and Nutrition, Hoseo University, Asan 31499, Republic of Korea; mhkang@hoseo.edu; 6Division of Animal Bioscience, School of Animal Life Convergence Sciences, Hankyong National University, Anseong 17579, Republic of Korea

**Keywords:** recombinant eCG, cAMP signaling, CHO-DG44 cells, phospho-ERK1/2, β-arrestin 2 recruitment

## Abstract

Equine chorionic gonadotropin (eCG) hormone, comprising highly glycosylated α- and β-subunits, elicits responses similar to follicle-stimulating hormone (FSH) and luteinizing hormone (LH) in non-equid species. This study aimed to establish a mass production of recombinant eCG (rec-eCG) using CHO DG44 cells. Single-chain rec-eCG β/α was expressed in CHO DG44 cells. FSH- and LH-like activities were evaluated in CHO-K1 and HEK 293 cells expressing the equine LH/CG receptor (eLH/CGR), rat LH/CGR (rLH/CGR), and rFSHR. pERK1/2 activation and β-arrestin 2 recruitment were assessed in PathHunter CHO-K1 cells. The expression from one, among nine isolates, peaked at 364–470 IU/mL on days 9 and 11. The molecular weight of rec-eCG β/α ranged from 40 to 47 kDa, with two distinct bands. PNGase F treatment reduced the molecular weight by 8–10 kDa, indicating N-glycosylation. Rec-eCG β/α demonstrated dose-responsive cAMP activity in cells expressing eLH/CGR, with enhanced potency in rLH/CGR and rFSHR. Phospho-ERK1/2 activation peaked at 5 min before declining rapidly. β-arrestin 2 recruitment was receptor-mediated in cells expressing hFSHR and hLH/CGR. This study provides insights into the mechanisms underlying eCG’s FSH- and LH-like activities. Stable CHO DG44 cells can produce large quantities of rec-eCG. eCG activates pERK1/2 signaling via the PKA/cAMP pathway and facilitates β-arrestin 2 recruitment.

## 1. Introduction

The equine chorionic gonadotropin (eCG) belongs to the family of glycoprotein hormones, members of which include luteinizing hormone (LH), follicle-stimulating hormone (FSH), and thyroid-stimulating hormone (TSH). It is a unique gonadotropic hormone with dual LH- and FSH-like activities in non-equid species [1,2,3,4]. Chorionic gonadotropins (CGs), produced by the placenta of primates and equids during early pregnancy [5] are composed of non-covalently connected α- and β-subunits. While the α-subunit is identical within the same species, the β-subunit has a unique structure that confers specific biological functions [6,7]. The equine chorionic girdle is composed of specialized invasive trophoblast cells that start forming around 25 days post-ovulation and differentiate into endometrial cups which detach from the chorionic girdle of the conceptus between days 37 and 120 of pregnancy [8,9,10]. The endometrial cups secrete eCG, which peaks at 70–80 days [11,12,13,14].

The C-terminal peptide (CTP) of the eCG β-subunit is heavily modified with multiple O-linked glycosylation sites, which significantly extend its circulatory half-life and enhance secretion in mammalian cells [15,16,17]. Interestingly, despite differences in their origin (placenta for eCG and pituitary for eLH), the β-subunits of eCG and eLH share the same structure [18,19]. Natural eCG has been widely used in domestic and experimental animals to induce superovulation and increase ovulation rates [20,21,22]. Often, eCG is combined with human chorionic gonadotropin (hCG) to regulate ovarian function and induce ovulation [6,23]. Recombinant eCG (rec-eCG) exhibits both LH-like and FSH-like activities in primary rat Leydig and granulosa cells [7], as well as in rat LH/CG receptors (rLH/CGR) and FSH receptors (rFSHR) [24]. Furthermore, rec-eCG induces ovulation *in vivo* [6]. However, a large-scale production of rec-eCG from Chinese hamster ovaries (CHO)-K1 and CHO-suspension (CHO-S) cells has proven challenging. Recent advances have demonstrated the potential for stable, high-yield expressions of glycoproteins in CHO-DG44 cells, including human FSH [25], hCG [26], anti-TNF alpha antibodies [27], and eel LH and FSH [28].

Our previous work showed that recombinant eel LH and FSH produced in CHO-DG44 cells exhibited 10–14-fold higher secretion levels compared to transient expression systems using CHO-K1 and CHO-S cells [29]. The LH/CGR belongs to the G protein-coupled receptor (GPCR) superfamily [30,31], and it mediates the signaling cascade that includes cAMP accumulation and extracellular signal-regulated kinase (ERK1/2), which is activated by phosphorylation (pERK1/2) [32,33]. Both Gαs proteins and β-arrestins contribute to ERK signaling through G protein-dependent and β-arrestin 2-dependent mechanisms, respectively [34,35,36,37,38]. Additionally, β-arrestins and GPCR kinases (GRKs) are crucial in FSHR signaling [39,40]. siRNA-mediated β-arrestin1/2 knockdown revealed reduced FSHR-stimulated pERK1/2 activation, while CRISPR-based β-arrestin1/2 knockout and subsequent reintroduction enhanced the activation [41]. However, the mechanism of eLH/CGR-mediated pERK1/2 activation and its associated signaling pathways remains unknown.

Currently, eCG is commercially obtained through repeated blood collection from pregnant mares, a practice that raises significant animal welfare concerns. rec-eCG has been shown to exhibit full biological activity both in vitro and in vivo [13,24]. However, the secretion levels in mammalian cell systems remain insufficient for practical field applications. Therefore, the development of a mass production system for rec-eCG in mammalian cells is essential.

In this study, we report on the characterization of rec-eCG produced in CHO-DG44 cells. Our findings demonstrate that a single-chain form of eCG exhibits full biological activity and induces pERK1/2 activation in vitro in cells expressing rLH/CGR, rFSHR, and eLH/CGR. These results highlight the potential for a large-scale production of rec-eCG with LH-like and FSH-like activities using CHO-DG44 cells.

## 2. Materials and Methods

### 2.1. Materials

Oligonucleotides were synthesized by Genotech (Daejeon, Republic of Korea). Disposable spinner flasks and spinner flasks with magnetic stirring bars were acquired from Corning Inc. (Corning, NY, USA). RIPA buffer and other miscellaneous reagents were obtained from Merck Sigma-Aldrich Corp. (St. Louis, MO, USA).

### 2.2. Cells and Media

The Freedom™ DG44 Kit, CHO DG44 cells, CD DG44 medium, CD OptiCHO™ me-dium, CD FortiCHO™ medium, cloning medium, Lipofectamine 2000, and antibiotics were purchased from Invitrogen Corporation (Carlsbad, CA, USA). CHO-K1 cells and HEK 293 cells were sourced from the Korean Cell Line Bank (Seoul, Republic of Korea). Ham’s F-12 medium, fetal bovine serum (FBS), CHO-S-SFMII medium, and OptiMEM medium were purchased from Gibco BRL (Grand Island, NY, USA).

### 2.3. Construction of Single-Chain eCGβ/α

A single-chain eCG was constructed by fusing the full-length eCG β-subunit cDNA with the α-subunit cDNA using overlapping PCR mutagenesis [6]. Additionally, a myc-tag (10 amino acids: Glu-Gln-Lys-Leu-Ile-Ser-Glu-Glu-Asp-Leu) was inserted in the mature protein between the first two amino acids of the eCG β-subunit [24]. The resulting PCR fragments encoding the single-chain eCG were ligated into the pGEMT-Easy vector (Promega, Madison, WI, USA) using the DNA ligation kit (Takara Bio, Shiga, Japan). Subsequently, the full-length eCG fragments were inserted into the pOptiVEC TOPO TA Cloning expression vector using a kit (Invitrogen Corporation, Carlsbad, CA, USA) and following the manufacturer’s instructions. The presence of the Kozak sequence and myc-tag was confirmed, and any PCR-induced errors were identified by sequencing the plasmids. The orientation of the insertion was verified using restriction enzyme (Takara Bio, Shiga, Japan) digestion. A schematic diagram of the single-chain eCG construction is shown in Figure 1.

### 2.4. Transfection into CHO DG44 Cells and Isolation of Single Cells Expressing rec-eCG Proteins

The expression vectors were linearized using the PvuI restriction enzyme (Takara Bio, Shiga, Japan) and transfected into CHO DG44 cells with the FreeStyle™ MAX reagent (Invitrogen Corporation, Carlsbad, CA, USA), as previously described [29]. Briefly, CHO DG44 cells were seeded at a density of 3 × 10^5^ cells/mL one day prior to transfection. On the day of transfection, the cell density was approximately 5 × 10^5^ cells/mL. Plasmid DNA (18 µg) was isolated using QIAprep-Spin plasmid kit following manufacturer’ protocol (Qiagen Inc., Hilden, Germany), diluted in 600 µL of OptiPRO™ serum-free medium (SFM), while 15 µL of FreeStyle™ MAX reagent was diluted in a separate 600 µL of OptiPRO™ SFM. The DNA and reagent solutions were gently mixed by inversion and incubated for 10 min to form DNA–FreeStyle™ MAX complexes. These complexes were then added dropwise to the cells while gently swirling the flask.

At 48 h post-transfection, cells were collected via centrifugation at 300× *g* for 5 min and were cultured in a complete CD OptiCHO™ medium supplemented with 8 mM L-glutamine for the first round of selection [33,34]. The medium was replaced every 3–4 days with 30 mL of fresh medium until cell viability exceeded 90%, which typically took 2–3 weeks. To amplify the introduced gene by inhibiting dihydrofolate reductase (dhfr), cells were subjected to MTX (Invitrogen Corporation, Carlsbad, CA, USA) treatment. Initially, 500 nM MTX was applied for 3 weeks, followed by subsequent rounds of treatment with 2 µM and 4 µM MTX to enhance the integration locus. The resulting cell pools were aliquoted and frozen at −80 °C. The conditioned medium from these cultures was sterilized via membrane filtration, aliquoted, and stored at −80 °C until further use. To perform limiting dilution cloning, amplified cells were expanded in medium without glutamine for two passages. Cells were then serially diluted to a density of 0.5–2 cells/100 µL in complete cloning medium and dispensed into 96-well plates (100 µL per well). The growth of monoclonal colonies was monitored under a microscope. Selected single cells were transferred to 24-well plates containing fresh growth medium supplemented with 6 mM L-glutamine, followed by an expansion in the 6-well plates and T-25 flasks. Finally, selected clones were aliquoted and stored at −80 °C. For further analysis, individual clones were cultured in 125 mL flasks at 37 °C with 8% CO_2_ and shaken at 130–135 rpm.

### 2.5. Production and Quantitation of rec-eCG Proteins

To evaluate rec-eCG protein production, selected cells were seeded at a density of 3 × 10^5^ viable cells/mL in 30 mL of fresh medium. The culture medium samples (2 mL) were collected on days 0, 1, 3, 5, 7, 9, and 11. The collected supernatants were centrifuged at 100,000× *g* for 10 min at 4 °C to remove cell debris. A portion of the supernatant was concentrated using a Centricon filter (Centriplus centrifugal filter devices, Amicon Bioseparations, Billerica, MA, USA) for subsequent PMSG enzyme-linked immunosorbent assay (ELISA) analysis using PMSG ELISA kit (DRG International Inc., Mountainside, NJ, USA) and cAMP assays. Quantification of rec-eCG proteins was performed using a PMSG ELISA with an anti-PMSG monoclonal antibody, horseradish peroxidase (HRP)-conjugated secondary antibody, and TMB substrate, following the manufacturer’s protocol. In brief, 100 µL of raw culture medium or a 40–500-fold diluted sample was dispensed into 96-well plates pre-coated with the monoclonal antibody and incubated at room temperature for 60 min without agitation. Plates were washed thrice with distilled water, 100 µL of HRP-conjugated secondary antibody was added to each well, and the plates were incubated for another 60 min without agitation. After washing the plates five times with 300 µL of distilled water, 100 µL of TMB substrate solution was added, incubated for 30 min, and the enzymatic reaction was stopped by adding 50 µL of stop solution to each well. Absorbance at 450 nm was measured within 30 min using a plate reader. Based on the assay protocol’s conversion factor, 1 IU was equated to 100 ng.

### 2.6. Western Blotting and Enzymatic Digestion of N-Linked Oligosaccharides

For Western blot analysis, 20 µL of the collected supernatant was analyzed using reducing sodium dodecyl sulfate-polyacrylamide gel electrophoresis (SDS-PAGE). The separated proteins were transferred (Bio-Rad Mini Trans-Blot system) onto a 0.2 µm polyvinylidene difluoride (PVDF) membrane, washed with Tris-buffered saline containing Tween 20 (TBS-T), and incubated with the primary anti-myc antibody (1:5000 dilution; Invitrogen Corporation, Carlsbad, CA, USA). Further incubation was performed in the presence of the secondary HRP-conjugated goat anti-mouse IgG antibody (1:3000 dilution). Subsequently, the membrane was treated with 2 mL of Lumi-Light substrate solution for 1 min (Lumi-Light Western blot kit, Roche, Pleasanton, CA, USA) and exposed to X-ray films for 1–10 min. To analyze the glycosylation of rec-eCG proteins, N-linked glycans were enzymatically removed using the deglycosylation kit PNGase F (New England Biolabs, Ipswich, MA, USA) following the manufacturer’s instructions. Briefly, 20 µg of rec-eCG proteins were incubated with PNGase F, reaction buffer, and NP-40, for 1 h at 37 °C. The samples were subjected to SDS-PAGE, followed by Western blot to analyze the removal of N-linked glycans.

### 2.7. Construction of eLH/CGR, rLH/CGR, and rFSHR Expression Vectors

The eLH/CGR, rLH/CGR, and rFSHR genes were cloned into the mammalian expression vector pcDNA3 (Invitrogen Corporation, Carlsbad, CA, USA) and pCORON 1000SP VSV-G tag expression vectors (Amersham Biosciences, Piscataway, NJ, USA), as previously described [24]. PCR fragments of these receptor genes were inserted into the pcDNA3 vector at the *EcoRI* and *XhoI* restriction sites, resulting in constructs designated as pcDNA3-eLH/CGR, pcDNA3-rLH/CGR, and pcDNA3-rFSHR. Additionally, receptor cDNAs lacking signal sequences were subcloned into the eukaryotic expression vector pCORON 1000SP VSV-G tag, yielding constructs designated as pVSVG-eLH/CGR, pVSVG-rLH/CGR, and pVSVG-rFSHR.

### 2.8. cAMP Analysis Using Homogeneous Time-Resolved Förster Resonance Energy Transfer (HTRF) Assays

The total cAMP levels in cells expressing eLH/CGR, rLH/CGR, and rFSHR were quantified using cAMP Dynamics 2 competitive immunoassay kits (Cisbio Bioassays, Codolet, France). The assay utilized a cryptate-conjugated anti-cAMP monoclonal antibody and d2-labeled cAMP. Transfected cells harboring eLH/CGR, rLH/CGR, or rFSHR plasmids were seeded into 384-well plates at a density of 10,000 cells per well. To stimulate the cells, 5 µL of compound medium buffer containing rec-eCG was added to each well, followed by a 30 min incubation at room temperature. Subsequently, 5 µL of cAMP-d2 and 5 µL of anti-cAMP-cryptate were added to the wells and further incubated for 1 h. cAMP levels were measured using an Artemis K-101 HTRF microplate reader (Kyoritsu Radio, Minato-ku, Japan). The results were calculated as the ratio of fluorescence intensities at 665 nm and 620 nm and expressed as Delta F% (cAMP inhibition). Delta F% values were analyzed using GraphPad Prism software version 6.0 (GraphPad, Inc., La Jolla, CA, USA).

### 2.9. Measurement of pERK1/2 Levels by Homogeneous Time-Resolved Förster Resonance Energy Transfer (HTRF) Assays

Plasmids containing eLH/CGR, rLH/CGR, and rFSHR (pCORON1000 SP VSV-G) were transfected into HEK 293 cells. The cells were plated at a density of 1.5 × 10^4^ cells per 8 µL in a 384-well plate using HBSS medium, 48 h post-transfection. Cells were stimulated with rec-eCG at varying concentrations (50, 100, 250, and 500 ng/mL) for 7 min to analyze dose-dependent responses. Time-dependent responses were assessed by treating cells with 50 and 250 ng/mL of rec-eCG for 0 to 30 min. pERK1/2 activation was measured in cell lysates using an HTRF assay (Cisbio Phospho-ERK [Thr202/Tyr204] cellular kit, Codolet, France). Briefly, 4 µL of lysis buffer was added to the wells and incubated by shaking for 30 min. Further, 4 µL of premixed antibody solutions containing d2 acceptor and Eu^3+^-Cryptate donor-labeled antibodies were added and incubated at room temperature for 2–4 h. Total ERK1/2 levels were also measured to monitor steady-state protein levels and normalize phosphorylated ERK1/2 levels. The plates were read at two wavelengths (665 nm and 620 nm) using a TriStar2 S LB942 microplate reader. The results were calculated as a ratio using the formula: [Ratio% = (signal at 665 nm/signal at 620 nm) × 10^4^]. Rec-eCG-stimulated HTRF ratios were normalized to each experiment and expressed as the fold change compared to unstimulated cells.

### 2.10. Measurement of Phospho-ERK1/2 by Western Blot

HEK 293 cells were seeded in six-well plates and transfected with plasmids encoding eLH/CGR, rLH/CGR, and rFSHR (pCORON1000 SP VSV-G). The pERK1/2 assay was conducted 48 h after transfection. Before stimulation, cells were starved in serum-free medium for 4–6 h. Stimulation was carried out using recombinant eCG in a dose-dependent manner (0 to 2000 ng/mL) for eLH/CGR or for a time-dependent manner of up to 60 min for eLH/CGR and 30 min for rLH/CGR and rFSHR. Following stimulation, cells were lysed using RIPA buffer supplemented with a protease inhibitor cocktail. Protein concentrations were determined using the Bradford assay. Equal amounts of protein (20–40 µg) were separated on 10% SDS-PAGE gels and transferred to PVDF membranes. Membranes were incubated overnight at 4 °C with a polyclonal anti-phospho-ERK1/2 antibody (1:2000 dilution) and a monoclonal anti-ERK1/2 antibody (1:3000 dilution) (Cell Signaling Technology, Danvers, MA, USA). This was followed by incubation with HRP-conjugated anti-rabbit and anti-mouse secondary antibodies (Cell Signaling Technology, Beverly, MA, USA) for 1 h and detection by chemiluminescence using SuperSignal™ Western Pico reagent (Thermo Fisher Scientific Inc., Waltham, MA, USA). Densitometric quantification of the immunoblots was conducted using Image Lab v6.0 software (Bio-Rad, Hercules, CA, USA).

### 2.11. Measurement of β-Arrestin 2 Recruitment

β-arrestin 2 recruitment was assessed using enzyme fragment complementation with PathHunter (DiscoverX) eXpress CHO-K1 cells expressing hFSHR and hLH/CGR (Eurofins DiscoverX, Fremont, CA, USA). The cells were seeded at a density of 0.5 × 10^4^ cells per well in a 384-well plate and incubated at 37 °C for 24 or 48 h. The cells were then treated with 2200 ng/mL of rec-eCG agonist in a time-dependent manner. Additionally, varying concentrations of rec-eCG agonist were applied for 30 min. PathHunter detection reagents were prepared by combining 19 parts cell assay buffer, 5 parts substrate reagent 1 and 1 part substrate reagent 2. Each well received 13.75 µL of detection reagent, followed by a 60 min incubation at room temperature. Luminescence was measured using a plate reader.

### 2.12. Data and Statistical Analysis

Sequence data were analyzed using the Multalin multiple sequence alignment tool. Dose–response curves were generated from experiments performed in duplicate. cAMP levels were corrected by subtracting background signals obtained from mock-transfected cells. GraphPad Prism (v. 6.0, San Diego, CA, USA) was used for the analysis of cAMP activity, EC_50_ values, and stimulation curves. Each curve from a single experiment was normalized to the background signal recorded in mock-transfected cells. Phospho-ERK1/2 data were visualized using GraFit (v. 5.0, Erithacus Software, Horley, Surrey, UK). Results are expressed as mean ± standard error of the mean (SEM) from three independent experiments. Comparisons between multiple groups were performed using one-way analysis of variance (ANOVA) followed by Tukey’s post hoc test in GraphPad Prism. Differences were considered statistically significant at *p* < 0.05.

## 3. Results

### 3.1. Isolation of Single Cells Expressing rec-eCG in CHO-DG44 Cells

In the first round of selection, transfected cells were grown in the CD-OptiCHO medium for about three weeks. During the initial medium exchange after three days, cell viability dropped significantly to 39.4% and remained low after the second exchange. However, cells recovered from low viability, reaching 70.3% and eventually exceeding 85% during the subsequent medium exchange. Thereafter, viability consistently remained above 94%, and the selected cells were cryopreserved in a liquid nitrogen tank as a pooled cell population. In the second round of selection, methotrexate (MTX) concentration was gradually increased to amplify the transfected gene. Treatment with 500 nM MTX reduced cell viability to approximately 60% but viability recovered to over 90% within three weeks. When the MTX concentration increased to 2 μM, cell viability decreased to 70% but gradually returned to 90% within one month. Treatment with 4 μM MTX resulted in a slight decrease in viability but remained above 75%, ultimately recovering to over 93% after the third adjustment. Single-cell isolation was performed using 96-well plates and complete cloning medium. The initial appearance of single cells was observed under a microscope approximately 14 days post-isolation in 96-well plates. After three to four weeks, isolated single cells were transferred to 24-well plates. Ultimately, nine single-cell clones were successfully isolated using the complete cloning medium.

To quantify the secretion levels of rec-eCG in the culture medium, supernatants were collected on days 1, 3, 5, 7, 9, and 11 post-culture. As shown in Figure 2, rec-eCG concentrations gradually increased over the cultivation period. On day 3, secretion levels were relatively low, ranging from 35 to 56 IU/mL. By day 5, concentrations increased to between 100 and 199 IU/mL. Clone No. 4 exhibited high expression levels, with concentrations reaching 310 ± 5 IU/mL on day 7. By day 9, secretion levels exceeded 350 IU/mL across all clones. Notably, clone No. 9 demonstrated the highest secretion levels, reaching 464 ± 29 IU/mL on day 9 and 470 ± 24 IU/mL on day 11. The secretion levels of rec-eCG remained consistently high across all clones through day 11. These results show that the expression of rec-eCG increased steadily over the culture period, with optimal secretion observed on days 9 and 11. Thus, CHO-DG44 cells were shown to be capable of producing rec-eCG in large quantities.

### 3.2. Western Blot Analysis of rec-eCG

The supernatant media from the final cell clones were collected on days 7 and 9 post-cultivation. Twenty microliters of the medium were subjected to Western blot analysis. Distinct bands were observed in all samples within a broad molecular weight range of 40–47 kDa (Figure 3). This indicates that the molecular weight of rec-eCG produced by CHO-DG44 cells falls between 40 and 47 kDa, with two specific bands clearly visible in the results. To further analyze secretion dynamics, we examined the quantities of rec-eCG produced over various culture days using four clones (No. 1, 2, 3, and 4). Broad bands were detected across all clones (Figure 4). Weak bands appeared on day 3, and the band intensity gradually increased, peaking on days 7 and 9, before slightly declining on day 11. These observations align with ELISA data, which reflected the secretion profile of rec-eCG over time. The PNGase treatment resulted in a marked decrease in rec-eCG molecular weight by approximately 8–10 kDa, giving rise to bands of about 32–37 kDa (Figure 5). These findings suggest that rec-eCG undergoes appropriate post-translational oligosaccharide modifications in CHO-DG44 cells. Thus, CHO-DG44 cells are a suitable host for producing heavily glycosylated recombinant glycoproteins.

### 3.3. cAMP Responsiveness of rec-eCG in Cells Expressing eLH/CGR, rLH/CGR, and rFSHR

The in vitro LH-like activity of recombinant equine chorionic gonadotropin (rec-eCG) was evaluated using CHO-K1 cells expressing eLH/CGR, rLH/CGR, and rFSHR. The capacity of rec-eCG to stimulate cAMP production is shown in Figure 6. Rec-eCG elicited a dose-dependent increase in cAMP levels in cells expressing both eLH/CGR and rLH/CGR. The half-maximal effective concentration (EC_50_) and maximal response (Rmax) values in eLH/CGR-expressing cells were 0.2 ng/mL and 186.8 ± 3.1 nM/10^4^ cells, respectively (Table 1). For rLH/CGR-expressing cells, these values were 0.03 ng/mL and 85.5 ± 1.4 nM/10^4^ cells, respectively. The dose–response curve for rLH/CGR shifted slightly to the left, with an EC_50_ value 6.6-fold lower than that of eLH/CGR. However, the Rmax value for rLH/CGR was approximately 0.46-fold lower than that of eLH/CGR. These findings indicate that rec-eCG exhibits higher Rmax responsiveness for cAMP production in eLH/CGR-expressing cells compared to rLH/CGR-expressing cells, demonstrating potent LH-like biological activity in both receptor types. The in vitro FSH-like activity of rec-eCG was assessed using CHO-K1 cells expressing rFSHR. Rec-eCG also induced a dose-dependent increase in cAMP levels in rFSHR-expressing cells, following a pattern similar to that observed in rLH/CGR-expressing cells (Figure 6C). The EC_50_ and Rmax values for FSH-like activity were 0.1 ng/mL and 50.3 ± 0.9 nM/10^4^ cells, respectively. The dose–response curve for rFSHR shifted further to the right compared to rLH/CGR. The EC_50_ value for rFSHR was approximately 2-fold higher than that of eLH/CGR, while the Rmax value was only 0.27-fold of that observed for eLH/CGR. These EC_50_ values in the rLH/CGR and rFSHR are consistent with the results observed in the CHO-S cells [24].

These results suggest that cAMP responsiveness in rLH/CGR-expressing cells is more robust than in rFSHR-expressing cells. Overall, these findings demonstrate that rec-eCG possesses dual LH-like and FSH-like activities in cells expressing the respective receptors. Although the Rmax values in rLH/CGR- and rFSHR-expressing cells were 0.46- and 0.27-fold lower, respectively, compared to eLH/CGR, rec-eCG effectively engages both rLH/CGR and rFSHR-mediated cAMP signaling pathways. This study highlights the signal transduction properties of rec-eCG in cells expressing eLH/CGR, rLH/CGR, and rFSHR and confirms its potent biological activity in non-equid species. Together our results provide insights into the dual biological activity of rec-eCG, which could be used to form strategies for regulating its activity in applied and experimental settings.

### 3.4. Identification of eLH/CGR-, rLH/CGR-, and rFSHR-Mediated pERK1/2 Activation

To investigate whether rec-eCG is involved in the pERK1/2 signaling pathway, we assessed its ability to activate the pERK1/2 cascade in HEK 293 cells. Initially, pERK1/2 responsiveness was measured using the homogenous time-resolved Förster resonance energy transfer (HTRF) method. The results revealed that rec-eCG induced dose-dependent pERK1/2 activation within 7 min of agonist treatment in cells expressing eLH/CGR (Figure 7A). Next, we analyzed the time-dependent activation of pERK1/2 at two different concentrations of rec-eCG (50 ng/mL and 250 ng/mL) (Figure 7B,C). At 5 min, stimulation with 50 ng/mL rec-eCG increased pERK1/2 activation by 2.6-fold relative to basal activity, which remained stable until 15 min. Treatment with 250 ng/mL of rec-eCG resulted in a robust 7.8-fold increase in pERK1/2 activation, which was sustained up to 30 min. The highest levels of activation were observed at 5–10 min, with significant activation persisting after 30 min of exposure to 250 ng/mL rec-eCG. We further evaluated pERK1/2 activation in cells expressing rLH/CGR and rFSHR. As expected, rec-eCG stimulation at 50 ng/mL induced a modest 1.3–1.4-fold increase in pERK1/2 activation at 5–10 min, followed by a sharp decline (Figure 8A,B). At 250 ng/mL, pERK1/2 activation increased approximately 2.1–2.2-fold but also declined rapidly within 30 min (Figure 8C and Figure 9).

Notably, in cells expressing eLH/CGR, pERK1/2 activation was sustained for longer durations even at high agonist concentrations, unlike the rapid decline observed in rLH/CGR and rFSHR-expressing cells. These findings suggest that rec-eCG-mediated pERK1/2 activation operates through LH/CGR- and FSHR-dependent signaling pathways, with significantly higher activation levels in eLH/CGR-expressing cells compared to rLH/CGR and rFSHR-expressing cells.

To validate these observations, we confirmed pERK1/2 activation using Western blot analysis. Consistent with previous results, transient overexpression of eLH/CGR in HEK 293 cells demonstrated robust pERK1/2 activation upon rec-eCG stimulation. A sharp increase in activation was observed at 125 ng/mL rec-eCG, with activation reaching its peak at 250 ng/mL (Figure 10A). Interestingly, the activation levels remained elevated, with over 86% of peak activity sustaining at the highest concentration. In time-dependent experiments, pERK1/2 activation peaked at 5 min, decreased to 60% by 15 min, and maintained approximately 50% activity upto 60 min of rec-eCG exposure (Figure 10B).

The activation patterns in cells expressing rLH/CGR and rFSHR were similar to those observed in eLH/CGR-expressing cells, with maximum activation occurring at 5 min (Figure 11A,B). However, in these cells, activation declined rapidly to 10–20% of the maximum level within 15 min. The ERK1/2 cascade, a well-known mitogenic signaling pathway, was found to be responsive to eLH/CGR, rLH/CGR, and rFSHR activation upon rec-eCG treatment. Overall, our results demonstrate that rec-eCG induces pERK1/2 activation via eLH/CGR-, rLH/CGR-, and rFSHR-mediated signaling pathways. Moreover, pERK1/2 activation patterns paralleled cAMP responsiveness in these pathways, suggesting that rec-eCG-mediated pERK1/2 activation is linked to a Gs protein-dependent cAMP signaling cascade.

### 3.5. β-Arrestin 2 Recruitment in PathHunter CHO-K1 Cells

The potency of β-arrestin 2 recruitment was evaluated using enzyme fragment complementation in PathHunter eXpress CHO-K1 cells expressing human FSH receptors (hFSHR) and human LH/CG receptors (hLH/CGR). In hFSHR-expressing cells, β-arrestin 2 recruitment reached approximately 1.24-fold of the pretreatment level within 10 min of agonist treatment (Figure 12A). This recruitment increased to 1.44-fold after 20 min and remained consistent up to 90 min. The recruitment of β-arrestin 2 in hFSHR cells was dose-dependent, as shown in Figure 12B. In contrast, cells expressing hLH/CGR exhibited a much lower basal signal for β-arrestin 2 recruitment, approximately 10-fold lower than hFSHR cells. Recruitment increased by 1.34-fold within 10 min of agonist treatment and showed a slight further increase by 20 min (Figure 12C). Dose-dependent recruitment of β-arrestin 2 in hLH/CGR-expressing cells was minimal, as illustrated in Figure 12D. These findings suggest that the low responsiveness in hLH/CGR cells may be attributed to low receptor expression in the PathHunter CHO-K1 cell line. It is essential to recognize that the sensitivity of rec-eCG was systematically evaluated to determine whether the receptor expression level was the primary issue. This evaluation involved comparing the β-arrestin 2 recruitment data with cAMP responsiveness, which showed a nearly corresponding pattern. Overall, these results indicate that rec-eCG plays a significant role in activating eLH/CGR, rLH/CGR, and rFSHR through cAMP/PKA- and β-arrestin-biased signaling pathways.

## 4. Discussion

eCG is a glycoprotein hormone, composed of two subunits, and is secreted by the placenta of pregnant mares. It is produced by invasive trophoblast cells and endometrial cups and has been widely used for superovulation in experimental and domestic animals through purified preparations of pregnant mare serum gonadotropin. However, the conventional method of obtaining eCG involves collection of blood from many pregnant mares annually, raising significant ethical concerns. To address these issues, efforts have been directed toward producing rec-eCG in mammalian cells. Unfortunately, previous attempts using CHO-K1 and CHO-S cells failed to produce sufficient quantities to meet the demands [6,7]. In this study, we successfully achieved a large-scale production of rec-eCG using CHO-DG44 cells. The produced rec-eCG demonstrated full biological activity, including cAMP responsiveness in cells expressing eLH/CGR, rFSHR, and rLH/CGR. Additionally, pERK1/2 activation was significantly increased within 5 min, and rec-eCG effectively recruited β-arrestin 2 through hFSHR- and hLH/CGR-mediated pathways in PathHunter cells.

The biological functions of glycoprotein hormones, such as CG, FSH, LH, and TSH, are strongly influenced by the glycosylation patterns of their sugar chains, as demonstrated by specific N-glycosyl removal [42]. A comparative study on gene expression, using glycosylation mutants, showed that glycosylation sites play crucial roles in the ovulation rate in mouse ovaries [7]. For eCG and hCG β-subunits, the C-terminal peptide (CTP) regions (114–149 amino acids for eCG and 114–145 amino acids for hCG) are essential for early expression and biological activity [17,24]. Therefore, the CTP regions of the eCG β-subunit are indispensable for producing rec-eCG molecules with full biological activity in vivo.

In this study, we isolated nine single-cell clones producing rec-eCG, with the α-subunit fused to the β-subunit CTP. The secreted quantity of rec-eCG from these clones gradually increased over 11 days, reaching peak concentrations of approximately 364–470 IU/mL. All clones exhibited high expression quality, surpassing the expression levels achieved through transient production systems using third-generation lentiviral vectors in suspension CHO-K1 cells [43]. These results align with recent studies on recombinant eel LH and FSH, which demonstrated the successful establishment of stable expression systems in CHO-DG44 cells. This consistency underscores the potential of CHO-DG44 cells as a platform for large-scale production of glycoprotein hormones [29].

The molecular weight of purified natural eCG exhibits a complex range of bands from 50 to 90 kDa; however, rec-eCG produced in CHO-K1 cells showed a distinct molecular weight of approximately 46 kDa on size exclusion chromatography (SEC)-HPLC [15]. We detected rec-eCG bands in the range of 40–47 kDa, consistent with previous findings of 40–46 kDa in CHO-K1 and CHO-S cells [6]. Specifically, two distinct bands at 40 and 47 kDa were observed, which were absent in transient expression systems [24]. The differences between natural and recombinant eCG molecules are attributed to variations in glycan structures, with natural eCG having the highest carbohydrate content (over 40%) among glycoprotein hormones [15,44]. Comparative analysis of sugar chains has shown that natural eCG contains almost twice the sialic acid content of rec-eCG (9.1 mol Neu5Ac/mol) [13]. The importance of glycosylation in gonadotropins is well-established, as enzymatic digestion or mutations at specific glycosylation sites significantly reduce their activity and half-life [42,45].

Our findings showed that peptide N-glycosidase F treatment reduced the molecular weight of rec-eCG by about 8–10 kDa. This suggests that rec-eCG produced in CHO-DG44 cells underwent proper post-translational modification, resulting in heavily glycosylated oligosaccharides. Despite its lower molecular weight compared to natural eCG, rec-eCG demonstrated excellent suitability for glycoprotein hormone production in CHO-DG44 cells. We could not confirm whether the single-chain rec-eCG was modified with O-linked glycosylation structures. However, O-glycosidase treatment of rec-eCG, obtained from CHO-K1 cells, did not show a decrease in molecular weight [24]. Therefore, it is presumed that single chain eCG is not modified with O-linked glycosylation. In the case of recombinant hCG (SB005, based on reference products Ovidrel^®^ and Ovitrelle^®^), mass analysis revealed molecular weights ranging from 44 to 67 kDa, with four confirmed O-linked glycosylation sites at Ser121, Ser127, Ser132, and Ser138 on the β-subunit, as identified through peptide mapping [26].

The in vitro biological activity of hCG and FSH mutants lacking glycosylation sites is 5–10 times lower than their wild-type counterparts [46,47]. In the present study, rec-eCG demonstrated full biological activity in cells expressing eLH/CGR, rLH/CGR, and rFSHR. These findings are consistent with earlier reports, which showed that heterodimeric rec-eCGα/β and single-chain rec-eCGβ/α exhibit both LH-like and FSH-like activities comparable to natural eCG [24]. The results highlight the critical role of glycosylation sites in determining the biological activity of rec-eCG. The EC_50_ values for rec-eCG in eLH/CGR-expressing cells were 2.0–6.6 times lower than those observed in cells expressing rFSHR and rLH/CGR. However, the Rmax levels were relatively low, at 0.27- and 0.46-fold, respectively. Despite the lower EC_50_ values, rec-eCG significantly enhanced cAMP responses in eLH/CGR-expressing cells, further supporting its dual FSH- and LH-like activity in various species.

pERK activation was thoroughly monitored in these cells in a dose-dependent manner. Specifically, pERK activation analyzed by HTRF demonstrated a remarkable enhancement of approximately 8-fold in cells expressing eLH/CGR, following high-concentration stimulation with rec-eCG, compared to a more modest 2–2.2-fold increase observed in rLH/CGR and rFSHR. These findings were consistent with the results obtained by Western blot analysis, aligning with the observed cAMP responsiveness. Although β-arrestins have not been traditionally proposed to directly interact with or induce pERK activation, our results indicate that rec-eCG indeed induces pERK activation and stimulates receptor-mediated β-arrestin 2 signaling. Notably, in LH/CGR and FSHR, pERK activation peaked within 5 min, consistent with previous studies [29]. The major phosphorylation site in the C-terminus of hFSHR, critical for β-arrestin recruitment, negatively regulates β-arrestin-dependent ERK activation [39]. Moreover, hFSHR exhibits β-arrestin1/2 binding distinct from other GPCRs, such as the β1-adrenergic receptor (β1AR) or vasopressin 2 receptor (V2R) [41]. Arrestins interact with various signaling proteins, including components of the ERK1/2 cascade such as cRaf1, MEK1/2, and ERK1/2 [48,49,50]. Interestingly, while β-arrestin 2 is essential for β2AR internalization, it is not indispensable for ERK activation, which involves signaling through Gαs and Gβγ subunits [51]. Furthermore, GRKs interact with GPCRs via β-arrestins, as demonstrated by the overexpression of specific GRKs in cell clones lacking GRK2/3/5/6, leading to receptor-β-arrestin1/2 complexes mediated by selective kinases [52]. Our findings demonstrate pERK1/2 activation and β-arrestin 2 recruitment through eLH/CGR, rLH/CGR, and rFSHR in response to rec-eCG stimulation. Further research is necessary to determine whether pERK1/2 activation is driven by G protein-dependent mechanisms or β-arrestin-biased signaling. To assess the relative contributions of these pathways, we are currently generating β-arrestin1, β-arrestin2, and GRK knockout cells using CRISPR technology.

## 5. Conclusions

In the present study, we established a robust production system for rec-eCG using CHO-DG44 cells. The rec-eCG exhibited potent biological activity, inducing cAMP responsiveness and strong pERK1/2 activation in cells expressing eLH/CGR, rLH/CGR, and rFSHR. In PathHunter cells expressing hFSHR and hLH/CGR, rec-eCG stimulated the dose- and time-dependent recruitment of β-arrestin 2. These findings highlight the dual role of rec-eCG in activating both cAMP/PKA- and β-arrestin-biased pERK1/2 signaling pathways via eLH/CGR, rLH/CGR, and rFSHR. The successful large-scale production of rec-eCG proteins paves the way for their use as promising ovulation-inducing agents in female animals. Furthermore, our findings demonstrate that the FSH- and LH-like dual activities of eCG molecules can be precisely regulated, providing clear insights into their role in cAMP/PKA- and β-arrestin-biased signaling.

## Figures and Tables

**Figure 1 biomolecules-15-00289-f001:**
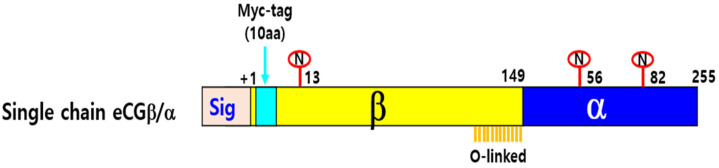
Schematic diagram of wild-type recombinant equine chorionic gonadotropin (rec-eCG). The diagram illustrates the N- and O-glycosylation sites on eCG. The eCG α-subunit has N-linked oligosaccharides at Asn56 and Asn82, while the β-subunit has one at Asn13. Additionally, the β-subunit includes up to 12 potential O-linked oligosaccharides in the carboxyl-terminal peptide (CTP) region. Circles labeled “N” and “O” indicate N-linked and O-linked glycosylation sites, respectively. A myc-tag epitope was inserted between the first and second amino acid residues of the mature β-subunit.

**Figure 2 biomolecules-15-00289-f002:**
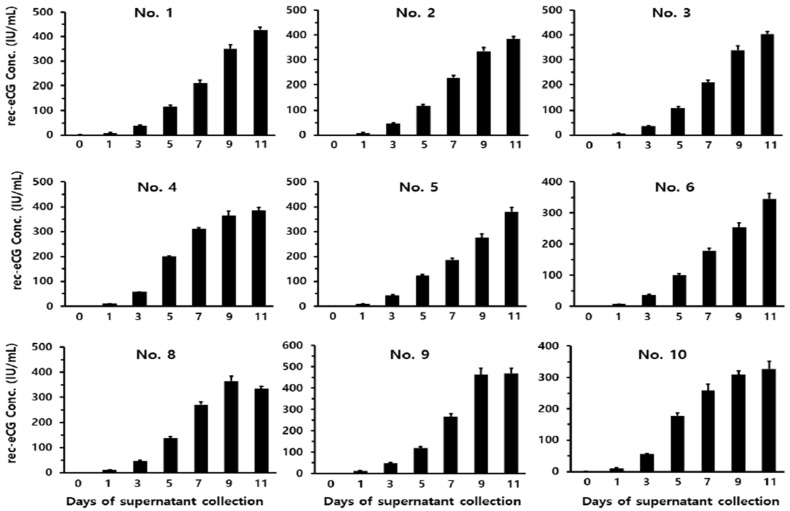
Quantitative analysis of rec-eCG production by ELISA following monoclonal cell isolation from CHO-DG44 cells. Nine monoclonal cell lines were isolated and evaluated for secreted rec-eCG levels. Supernatants were collected on days 0, 1, 3, 5, 7, 9, and 11 of culture in 50 mL spinner flasks. The expression levels of rec-eCG from each clone were analyzed using a sandwich enzyme-linked immunosorbent assay (ELISA). Data are presented as the mean ± standard error of the mean (SEM) from at least three independent experiments.

**Figure 3 biomolecules-15-00289-f003:**
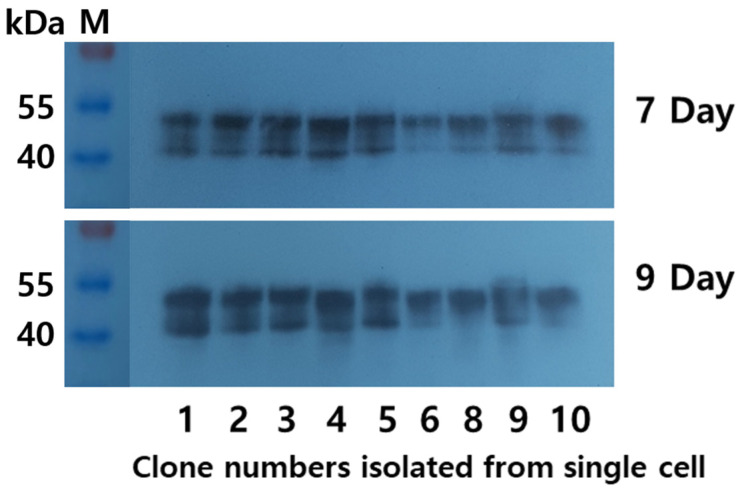
Western blot analysis of rec-eCG proteins produced by monoclonal cells. Supernatants from nine colonies were collected on days 7 and 9 of cultivation. Rec-eCG samples (20 µL) were resolved by sodium dodecyl sulfate-polyacrylamide gel electrophoresis (SDS-PAGE) and transferred to a membrane. Proteins were detected using anti-myc-tag antibodies and horseradish peroxidase-conjugated goat anti-mouse IgG antibodies. Original images can be found in Appendix A.

**Figure 4 biomolecules-15-00289-f004:**
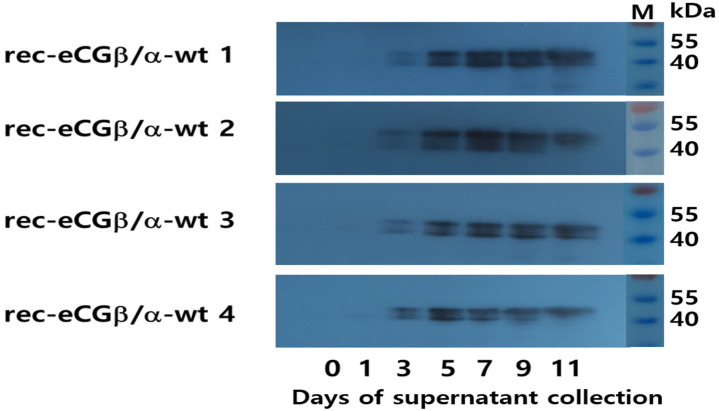
Western blot analysis of rec-eCG proteins over the cultivation period. Supernatants (20 µL) from four selected colonies were subjected to SDS-PAGE. Faint protein bands were first detected on day 3, with signal intensity gradually increasing over time. Two specific bands were consistently observed across all samples. Original images can be found in Appendix A.

**Figure 5 biomolecules-15-00289-f005:**
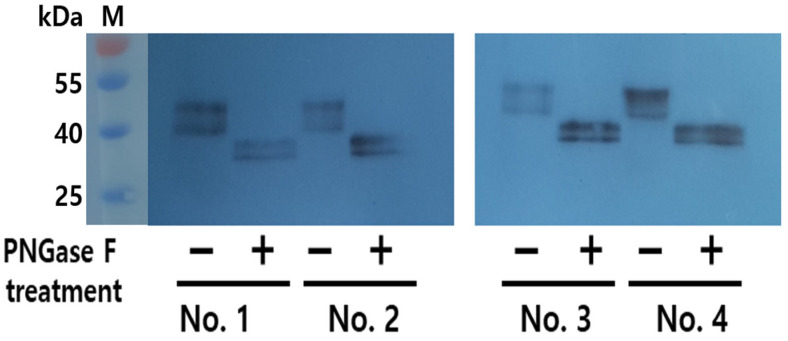
Deglycosylation analysis of rec-eCG proteins. Conditioned media from cells were treated with peptide-N-glycanase F (PNGase F) to remove N-linked oligosaccharides. Supernatants from cells No. 1 to 4 reacted with PNGase F at 37 °C for 1 h and then analyzed by SDS-PAGE. − indicates samples not treated with PNGase F, while + indicates samples treated with PNGase F. Original images can be found in Appendix A.

**Figure 6 biomolecules-15-00289-f006:**
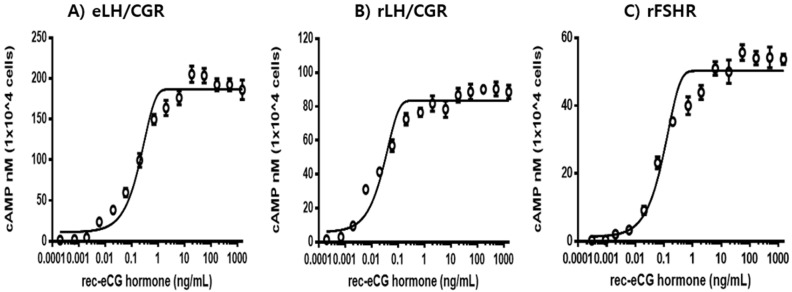
Effect of rec-eCG on cyclic AMP (cAMP) production in cells expressing equine LH/chorionic gonadotropin receptor (eLH/CGR), rat LH/CGR (rLH/CGR), and rat FSH receptor (rFSHR). Cells transiently transfected with eLH/CGR, rLH/CGR, or rFSHR were seeded in 384-well plates (10,000 cells/well) 24 h post-transfection. Cells were incubated with rec-eCG for 30 min at room temperature. cAMP production was measured using a homogeneous time-resolved fluorescence (HTRF) assay and expressed as Delta F%. The mock-transfected control values were subtracted from each dataset (see Methods). Data are shown as mean ± SEM from triplicate experiments, with curve fitting performed using a one-phase exponential decay model in GraphPad Prism. %. (**A**) eLH/CGR. (**B**) rLH/CGR. (**C**) rFSHR.

**Figure 7 biomolecules-15-00289-f007:**
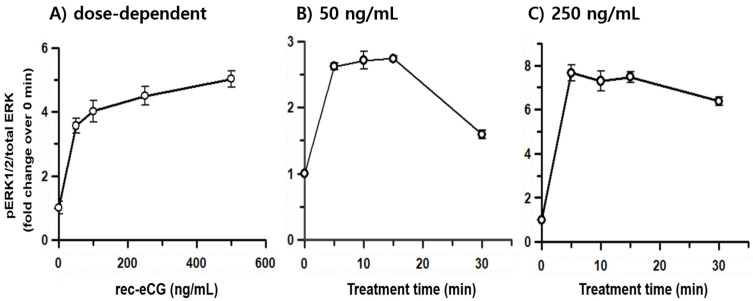
Dose- and time-dependent pERK1/2 activation by rec-eCG in cells expressing eLH/CGR. HEK293 cells transiently transfected with eLH/CGR were stimulated with rec-eCG under the following conditions: (**A**) Dose-dependent activation using 0, 50, 125, 250, and 500 ng/mL rec-eCG. (**B**) Time course of pERK1/2 activation with 50 ng/mL rec-eCG. (**C**) Time course of pERK1/2 activation with 250 ng/mL rec-eCG. Total ERK1/2 levels were assessed to normalize phosphorylated ERK1/2 (pERK1/2). Rec-eCG-stimulated HTRF ratios were normalized and expressed as fold changes relative to unstimulated cells.

**Figure 8 biomolecules-15-00289-f008:**
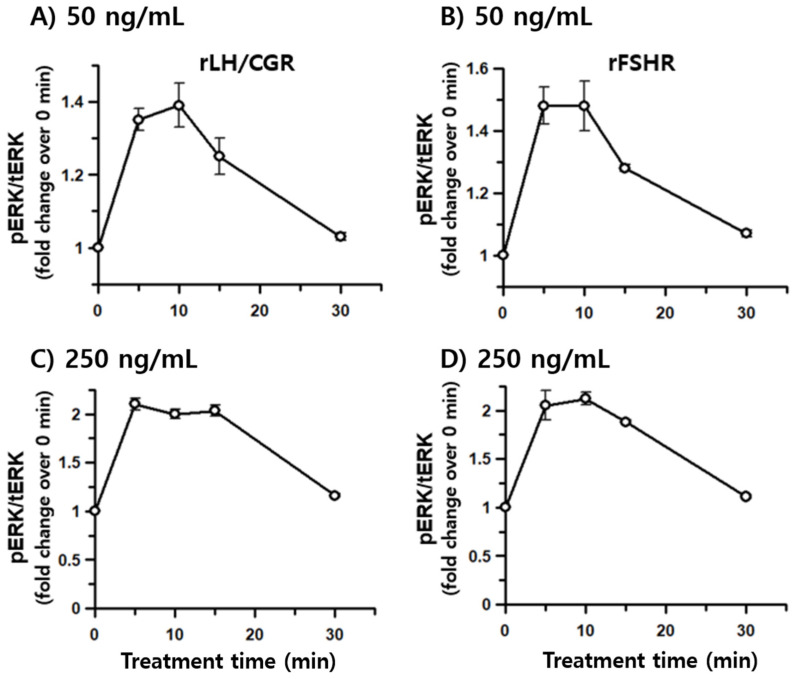
Dose- and time-dependent pERK1/2 activation by rec-eCG in cells expressing rLH/CGR and rFSHR. HEK293 cells transiently transfected with rLH/CGR or rFSHR were stimulated with rec-eCG under the following conditions: (**A**,**B**) pERK1/2 activation following treatment with 50 ng/mL rec-eCG. (**C**,**D**) pERK1/2 activation following treatment with 250 ng/mL rec-eCG.

**Figure 9 biomolecules-15-00289-f009:**
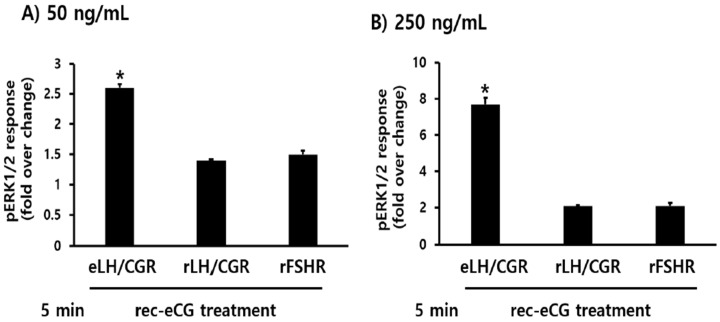
Comparison of pERK1/2 activation among eLH/CGR, rLH/CGR, and rFSHR. The pERK1/2 activation levels in eLH/CGR were compared with those in rLH/CGR and rFSHR at 5 min post-rec-eCG treatment. Data are presented as the mean ± standard error of the mean (SEM) from triplicate experiments. Values marked with asterisks indicate significant differences (* *p* < 0.05). (**A**) Activation at 50 ng/mL rec-eCG. (**B**) Activation at 250 ng/mL rec-eCG.

**Figure 10 biomolecules-15-00289-f010:**
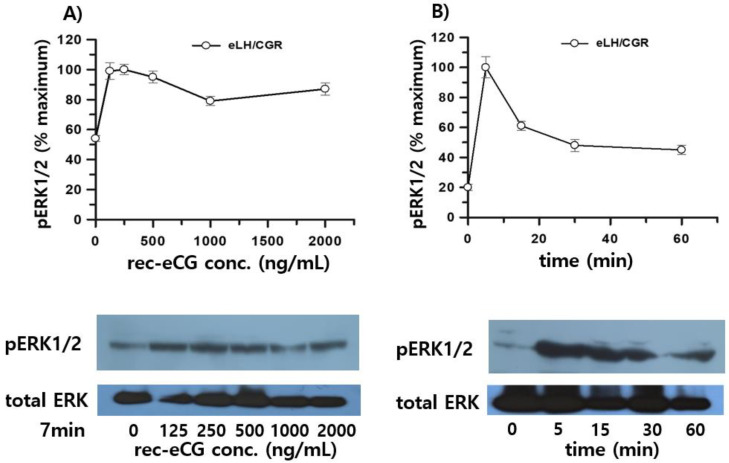
Effects of rec-eCG on pERK1/2 activation in eLH/CGR-stimulated cells. HEK293 cells transiently transfected with eLH/CGR were serum-starved for at least 6 h before stimulation. Cellular extracts (20 µg per sample) were analyzed by SDS-PAGE. (**A**) Dose-dependent pERK1/2 activation using rec-eCG concentrations of 0, 125, 250, 500, 1000, and 2000 ng/mL, to stimulate cells for 7 min. (**B**) Time course of pERK1/2 activation following treatment with 250 ng/mL rec-eCG. pERK1/2 and total ERK bands were quantified by densitometry, and pERK1/2 levels were normalized to total ERK levels. Equal protein amounts were loaded for each lane. Representative data are shown, and graphs depict the mean ± standard error (SE) from independent experiments. The maximal pERK1/2 response observed at 250 ng/mL and 5 min was designated as 100%. Original images can be found in Appendix A.

**Figure 11 biomolecules-15-00289-f011:**
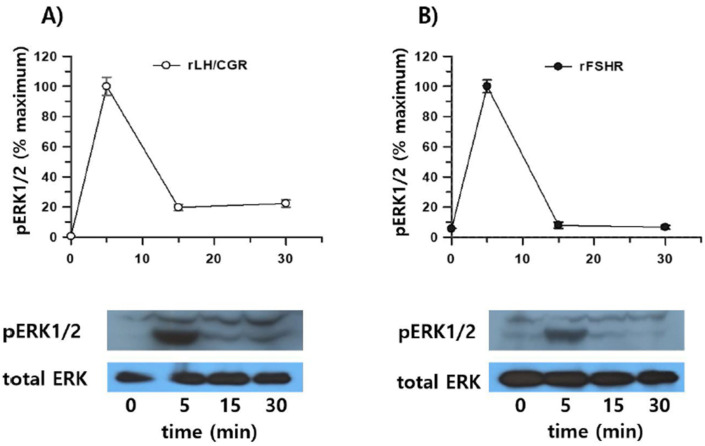
pERK1/2 activation stimulated by rLH/CGR and rFSHR. HEK293 cells transiently transfected with rLH/CGR or rFSHR were serum-starved for at least 6 h and stimulated with 250 ng/mL of agonist for the indicated times. Whole-cell lysates (20 µg per sample) were analyzed for pERK1/2 and total ERK levels by SDS-PAGE. pERK1/2 levels were normalized to total ERK levels. Representative data are shown, and graphs represent the mean ± SE from independent experiments. The maximal pERK1/2 response observed at 5 min was designated as 100%. (**A**) rLH/CGR. (**B**) rFSHR. Original images can be found in Appendix A.

**Figure 12 biomolecules-15-00289-f012:**
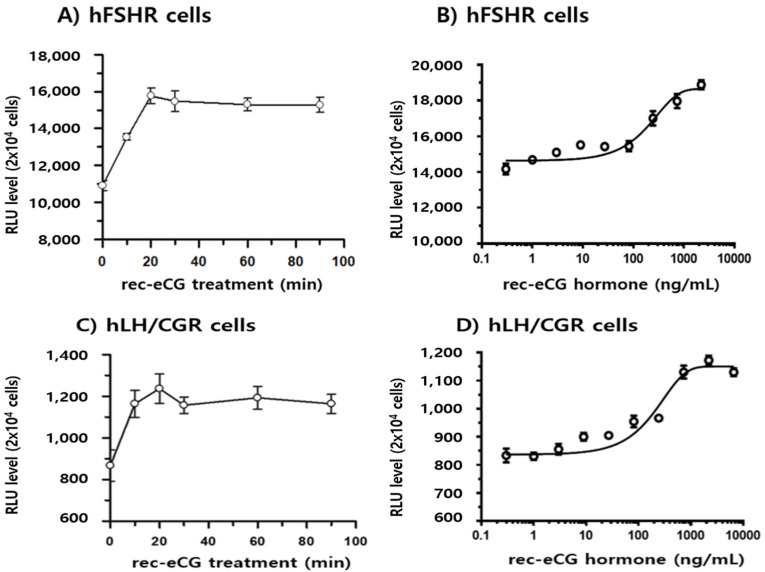
Effects of β-arrestin 2 recruitment in PathHunter (DiscoverX) eXpress CHO-K1 cells expressing hFSHR and hLH/CGR. Cells were plated at 0.5 × 10^4^ cells per well in 384-well plates and incubated for 24 or 48 h at 37 °C. Cells were stimulated with 2200 ng/mL of rec-eCG under dose- and time-dependent conditions. PathHunter detection reagents were added and incubated for 60 min at room temperature. Luminescence signals were measured using a plate reader. (**A**,**B**) β-arrestin 2 recruitment in CHO-K1 cells expressing hFSHR. (**C**,**D**) β-arrestin 2 recruitment in CHO-K1 cells expressing hLH/CGR.

**Table 1 biomolecules-15-00289-t001:** Bioactivity of rec-eCG in cells expressing eLH/CGR, rLH/CGR, and rFSHR.

Receptors	cAMP Responses		
	Basal *^a^* (nM/10^4^ Cells)	EC_50_ *^b^* (ng/mL)	Rmax *^c^* (nM/10^4^ Cells)
eLH/CGR	6.3 ± 0.9	0.20 (1.0-fold) (0.16 to 0.27) *^d^*	186.8 ± 3.1 (1.0-fold)
rLH/CGR	5.8 ± 0.6	0.03 (6.6-fold) (0.02 to 0.03)	85.5 ± 1.4 (0.46-fold)
rFSHR	1.3 ± 0.3	0.10 (2.0-fold) (0.08 to 0.13)	50.3 ± 0.9 (0.27-fold)

Data are presented as means ± standard error of the mean (SEM) from triplicate experiments. The half-maximal effective concentration (EC_50_) values were derived from concentration-response curves obtained in vitro. The cAMP responses for EC_50_ and Rmax are expressed relative to the eLH/CGR response, which was set as 1.0-fold. *^a^* Basal cAMP level: The average cAMP level in the absence of agonist treatment. *^b^* EC_50_: Half-maximal effective concentration. *^c^* Rmax: Maximum cAMP level per 10^4^ cells. *^d^* 95% confidence intervals.

## Data Availability

All data generated or analyzed during this study are included in this published article.

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
