# Peer review of "Enhanced Production and Functional Characterization of Recombinant Equine Chorionic Gonadotropin (rec-eCG) in CHO-DG44 Cells"

_biomolecules, 2025, doi:10.3390/biom15020289_

Round 1
Reviewer 1 Report
Comments and Suggestions for Authors
The present paper by M. Byambaragchaa et al deals with the production of single-chain equine chorionic gonadotropin (eCG) in CHO-DG44 cells and its in-vitro biological characterization in view of its use in place of natural eCG (PMSG) in reproduction control in farm animals.
General comments :
The aim of this study is interesting in the prospect of producing recombinant eCG in large quantities and at a limited cost to replace natural eCG. Indeed this eCG is obtained by bleeding pregnant mares to collect serum from which the gonadotropin is extracted and purified (PMSG).
For this, the authors have chosen the CHO-DG44 cells but have not buttressed their choice compared to other cell types; in particular, concerning the ability, or not, of these cells to achieve the O‑glycosylation of the eCG β-CTP region. Indeed this point is highly important to get an eCG with a long half-life comparable to that of PMSG (~2.5 days in blood circulation). This point should have been taken as crucial in selecting cells for the glycoprotein hormone expression.
In the different assays presented, a comparison with highly purified natural eCG (PMSG) would have been useful to ascertain the quality of the recombinant single-chain eCG.
Specific comments :
The absence of line numbering rendered the writing of these comments difficult.
#2.3 2nd § : MTX treatment : « … to enhance the integration locus. » is not clear.
#2.4 1st § : Sample concentration for eCG Elisa required ? Elisa not sensitive enough ? Afterward, the samples are diluted 40-500 fold.
#2.7 : It is not clear that intracellular cAMP is measured. It seems to be cAMP leaking from the cells. Or were the cells recovered from the medium and then broken off to release intra-cellular cAMP ?
#3.1 : Explain more how MTX exerts selection pressure on the cells.
#3.1 2nd § and fig 1: The quantities of eCG are given in IU. What is the standard preparation used in this Elisa ?
Given that the purest eCG preparations’ titer is ~10000 IU/mg the cells roughly produce ~50µg/mL equivalent pure natural eCG.
#3.2 p.8 : How can the sc-eCG MW after PNGase treatment be only 8-10kD ? It is less than the polypetide MW alone.
Fig.3 : The WB is performed only with the anti-myc tag. Why not also with anti-eCG antibodies ?
Fig.4 : Do the rec-eCG numbers in this figure correspond to the clone numbers in fig 2 ? If yes, why not show these selected molecules and not the nine products ? Is there no way to selectively remove O-glycosyl side chains? If not, it would be interesting to show that these bands represent O-glycosylated forms using specific lectins.
Fig.6 : It would have been interesting to express the eFSHR in these cells to confirm that eLH/CG indeed does bind/stimulate eFSHR and confirm that eFSH works in this system.
As mentioned above, it is not clear whether intra or extra or total cAMP was measured.
Fig.7 : Why « pre-treatment" ? what difference with treatment ? « time (min) » would be sufficient.
Fig.6, 7, 10, 12 : The sc-eCG concentrations are expressed in ng/mL. Have you desalted, freeze-dried and weighed the hormone ? Or are the concentrations determined by Elisa. If yes, how was the IU-µg conversion performed ?
Fig.10 : the caption is difficult to understand « …stimulated for 7 min under dose- and time-dependent conditions.
#3.5 : The percentages are difficult to follow. What is the « baseline » in fig. 12A ? 0 RLU or RLU level at time 0 ? Also, fig.11B does not correspond to β-arrestin as mentioned.
Fig. 12B and D : The abscissa should be interrupted to allow showing the RLU level at 0 eCG concentration.
#4 1st§ : The mares are not slaughtered but bled several times.
#4 2nd§ : Site-directed mutageneses of N-glycosyl sites are not fully convincing as they can induce conformational changes in the polypeptide chain. The experiments of N-glycosyl removal have also to be considered. To complete the last sentence add « in vivo » : « …. full in vivo biological activity ».
#4 4th§ : apparent molecular weight. Moreover, the sialic acid content in natural eCG is mainly borne by the O-glycosyl side chains in its βCTP. It would thus be important to get information concerning the CHO-DG44 to add the O-glycosyl side chains to the sc-eCG molecule and check their sialylation. It is an overstatement to write that « rec-eCG produced in CHO-DG44 cells underwent proper post-translational modification …. ».
Author Response
Comments and Suggestions for Authors
The present paper by M. Byambaragchaa et al deals with the production of single-chain equine chorionic gonadotropin (eCG) in CHO-DG44 cells and its in-vitro biological characterization in view of its use in place of natural eCG (PMSG) in reproduction control in farm animals.
General comments:
The aim of this study is interesting in the prospect of producing recombinant eCG in large quantities and at a limited cost to replace natural eCG. Indeed, this eCG is obtained by bleeding pregnant mares to collect serum from which the gonadotropin is extracted and purified (PMSG).
For this, the authors have chosen the CHO-DG44 cells but have not buttressed their choice compared to other cell types; in particular, concerning the ability, or not, of these cells to achieve the O‑glycosylation of the eCG β-CTP region. Indeed, this point is highly important to get an eCG with a long half-life comparable to that of PMSG (~2.5 days in blood circulation). This point should have been taken as crucial in selecting cells for the glycoprotein hormone expression.
In the different assays presented, a comparison with highly purified natural eCG (PMSG) would have been useful to ascertain the quality of the recombinant single-chain eCG.
Specific comments:
The absence of line numbering rendered the writing of these comments difficult.
#2.3 2nd § : MTX treatment : « … to enhance the integration locus. » is not clear.
→We changed “To amplify the introduction gene by inhibiting dihydrofolate reductase (DHFR) activity, in the 2.3.2nd.
#2.4 1st § : Sample concentration for eCG Elisa required ? Elisa not sensitive enough ? Afterward, the samples are diluted 40-500 fold.
→rec-eCG was subjected to the cAMP responsiveness at the receptor cells. ELISA system is very sensitive to rec-eCG indicating that R2 is 0.94 and samples were diluted with 40-500-fold according to the collected days as shown “ a 40-500-fold diluted sample” in the 2.4. It’s more than standard curves (0-800 mIU/mL).
#2.7 : It is not clear that intracellular cAMP is measured. It seems to be cAMP leaking from the cells. Or were the cells recovered from the medium and then broken off to release intra-cellular cAMP?
→We changed “intracellular cAMP” to “total cAMP” by reviewer’s comment in the 2.7.
#3.1 : Explain more how MTX exerts selection pressure on the cells.
→We also changed “these sentences” to “to amplify the integration gene” in the 3.1. 1st.
#3.1 2nd § and fig 1: The quantities of eCG are given in IU. What is the standard preparation used in this Elisa ?
Given that the purest eCG preparations’ titer is ~10000 IU/mg the cells roughly produce ~50µg/mL equivalent pure natural eCG.
→The PMSG enzyme-linked immunosorbent assay (ELISA) kit was purchased from DRG International Inc. (Mountainside, NJ, USA). Thus, we used the purified standard samples included in the Kit. The standard concentration is 0 to 800 mIU/mL. Therefore, the highest concentration on day 11 was the 40~50 ug/mL.
#3.2 p.8 : How can the sc-eCG MW after PNGase treatment be only 8-10kD ? It is less than the polypetide MW alone.
→We changed “the sentence” to “The PNGase treatment resulted in a marked decrease in rec-eCG molecular weight by approximately 8–10 kDa.” Thus, The molecular weight was decreased by approximately 8-10 kDa.
Fig.3 : The WB is performed only with the anti-myc tag. Why not also with anti-eCG antibodies?
→Previously, we produced eCG antibodies, but we didn’t get good results, so we don’t have eCG antibodies. Therefore, myc-tag was added to detect rec-eCG molecules with Western blotting.
Fig.4 : Do the rec-eCG numbers in this figure correspond to the clone numbers in fig 2 ? If yes, why not show these selected molecules and not the nine products? Is there no way to selectively remove O-glycosyl side chains? If not, it would be interesting to show that these bands represent O-glycosylated forms using specific lectins.
→Yes, the number in Fig. 4 is the same number in Fig. 2. We selected the 4 clone numbers representing a thick band by Fig. 2. We didn’t present the O-linked glycosylation elimination. We confirmed that single chain rec-eCG was to be unmodified the O-linked glycosylation chain in our previous studies using CHO-K1 and CHO-S cells. Perhaps this cause speculates that about a 12-linked sugar chain sites of eCG b-subunit were entered into the interior by folding post translation. Although it has not been announced yet, it has been confirmed that the O-linked glycosylation chains are modified by adding a specific base sequence to solve this problem.
Fig.6 : It would have been interesting to express the eFSHR in these cells to confirm that eLH/CG indeed does bind/stimulate eFSHR and confirm that eFSH works in this system.
As mentioned above, it is not clear whether intra or extra or total cAMP was measured.
→We analyzed the function of rec-eCG in the cells expressing rat FSHR, hFSHR and eel FSHR, but there are no binding results of re-eCG for eFSHR.
→As reviewer’s comment, we changed it to “total cAMP” as the same content of #2.7 comment.
Fig.7 : Why « pre-treatment" ? what difference with treatment ? « time (min) » would be sufficient.
→We changed “pre-treatment” to “treatment time (min)” in the Fig. 7 and Fig. 8.
Fig.6, 7, 10, 12 : The sc-eCG concentrations are expressed in ng/mL. Have you desalted, freeze-dried and weighed the hormone? Or are the concentration determined by Elisa. If yes, how was the IU-µg conversion performed?
→Yes, we concentrated by Centrifugal Filter Devices and Freeze-dried. After then, we reanalyzed by ELISA and 1 IU was assumed to be 100 ng according to the conversion factor of the suggested assay protocol.
Fig.10 : the caption is difficult to understand « …stimulated for 7 min under dose- and time-dependent conditions.
→We changed “for 7 min under dose- and time-dependent” to for 7 min under dose-dependent”. And the sentence moved to the (A).
#3.5 : The percentages are difficult to follow. What is the « baseline » in fig. 12A ? 0 RLU or RLU level at time 0 ? Also, fig.11B does not correspond to β-arrestin as mentioned.
→We changed “%” to “fold”.
→We changed “baseline’ to “pretreatment value”.
→RLU level at time 0 is correct and inserted RLU level in the Figure.
→We changed “Figure 11B” to “Figure 12B”.
Fig. 12B and D : The abscissa should be interrupted to allow showing the RLU level at 0 eCG concentration.
→We repeated this experiment many times. The highest concentration eCG (2200 ng/mL) was treated and the untreated sample of eCG was almost the same level at the lowest concentration sample (0.3 ng/mL).
→In GraphPad Prism software, even if the result of 0 is entered, the 0 result of the X-axis does not appear in the graph.
#4 1st§ : The mares are not slaughtered but bled several times.
→We changed “slughter” to “from the blood” in the #4 1st.
#4 2nd§ : Site-directed mutageneses of N-glycosyl sites are not fully convincing as they can induce conformational changes in the polypeptide chain. The experiments of N-glycosyl removal have also to be considered. To complete the last sentence add « in vivo » : « …. full in vivo biological activity».
→We changed “site-directed mutagenesis” to “specific N-glycosyl removal”.
→We inserted “in vivo”.
#4 4th§ : apparent molecular weight. Moreover, the sialic acid content in natural eCG is mainly borne by the O-glycosyl side chains in its βCTP. It would thus be important to get information concerning the CHO-DG44 to add the O-glycosyl side chains to the sc-eCG molecule and check their sialylation. It is an overstatement to write that « rec-eCG produced in CHO-DG44 cells underwent proper post-translational modification …. »
→We explained in Fig. 4 above.
→We confirmed that single chain rec-eCG was to be unmodified by the O-linked glycosylation chain in our previous studies using CHO-K1 and CHO-S cells. Perhaps this cause speculates that about a 12-linked sugar chain sites of eCG b-subunit were entered into the interior by folding post translation. Although it has not been announced yet, it has been confirmed that the O-linked glycosylation chains are modified by adding a specific base sequence to solve this problem.
→We inserted “Previous O-glycosidase results did not show a decrease in molecular weight. Therefore, it is presumed that single chain eCG is not modified with O-linked glycosylation because this part enters the inside in the folding process post-translation” in the #4 5th.

Reviewer 2 Report
Comments and Suggestions for Authors
The manuscript showed effective production of rec-eCG using CHO DG44 cells. Moreover, rec-eCG induces cAMP production, pERK1/2 signaling, and b-arrestin recruitment. I think that the manuscript is likely acceptable for Biomolecules. However, I hope that authors correct miner points as stated below.
Please add size directions in each Western Blotting data.
Please add information and references for EC50 values of rat LH - rLH/CGR and rat FSH - rFSHR in Results 3.3 or Discussion.
Author Response
The manuscript showed effective production of rec-eCG using CHO DG44 cells. Moreover, rec-eCG induces cAMP production, pERK1/2 signaling, and b-arrestin recruitment. I think that the manuscript is likely acceptable for Biomolecules. However, I hope that authors correct miner points as stated below.
Please add size directions in each Western Blotting data.
→We inserted marker size in Western Blotting data of Figure 3.4.5.
Please add information and references for EC50 values of rat LH - rLH/CGR and rat FSH - rFSHR in Results 3.3 or Discussi
→We inserted “These EC50 values in the rLH/CGR and rFSHR are consistent with the results produced in the CHO-S cells.

Round 2
Reviewer 1 Report
Comments and Suggestions for Authors
This second version of the paper by M. Byambaragchaa et al. is undoubtedly improved but still requires a few modifications to be acceptable for publication in Biomolecules.
The two points I would like the authors to address in the third (hopefully definitive) version are the following :
1/ O-glycosylation of rec eCG from DG44 cells
The authors indicate they failed to observe a decrease in eCG molecular weight upon O-glycosidase treatment. This might be due either to the absence of O-glycosylation in eCG expressed in CHO DG44 cells or to O-glycosidase inactivity.
To rule out O-glycosidase inactivity, a control measurement of its activity on natural PMSG could be provided. If this control is positive, this would suggest that CHO-DG44 cells do not achieve extensive O-glycosylation (as already found in other CHO cells). Like the authors, I do not think the O-glycosylated peptide (CTP) can be shielded from the medium. Indeed these sugars are highly hydrophilic and cannot be buried inside protein structures.
2/ Anti-eCG antibody
The authors indicate they possess no efficient anti-eCG antibody for their Western blot analysis. Why not use the one from the Elisa kit ?
I will be pleased to provide an anti-PMSG to the authors when this evaluation is over.
Author Response
Reviewer 1
This second version of the paper by M. Byambaragchaa et al. is undoubtedly improved but still requires a few modifications to be acceptable for publication in Biomolecules.
The two points I would like the authors to address in the third (hopefully definitive) version are the following :
1/ O-glycosylation of rec eCG from DG44 cells
The authors indicate they failed to observe a decrease in eCG molecular weight upon O-glycosidase treatment. This might be due either to the absence of O-glycosylation in eCG expressed in CHO DG44 cells or to O-glycosidase inactivity.
To rule out O-glycosidase inactivity, a control measurement of its activity on natural PMSG could be provided. If this control is positive, this would suggest that CHO-DG44 cells do not achieve extensive O-glycosylation (as already found in other CHO cells). Like the authors, I do not think the O-glycosylated peptide (CTP) can be shielded from the medium. Indeed these sugars are highly hydrophilic and cannot be buried inside protein structures.
→ Although these results are from CHO-K1 cells, a Western blot of the O-linked glycan cleavage results previously presented by our laboratory was provided. The N-linked cleavage showed a normal decrease in molecular weight, whereas the O-linked treatment did not result in any change in molecular weight. The construction of this gene presents the same cDNA nucleotide sequence as presented in this paper (Left).
→Therefore, some recent studies have reported that when glycoprotein hormones are produced in a dimer form rather than as a single chain, O-linked glycosylation occurs. Based on these findings, we considered how O-linked glycosylation is regulated and hypothesized that the linker region between the beta and alpha subunits in the single-chain form might be problematic. To address this, we introduced a specific nucleotide recognition sequence in this region, and as a result, we observed a decrease in molecular weight upon O-glycosidase treatment, confirming O-linked glycosylation. (Right).
→Therefore, this gene is currently being produced in DG44 cells. In conclusion, we speculate that during the post-translational folding process, the linker region between the beta and alpha subunits folds inward, preventing the modification of O-linked glycans.
2/ Anti-eCG antibody
The authors indicate they possess no efficient anti-eCG antibody for their Western blot analysis. Why not use the one from the Elisa kit ?
I will be pleased to provide an anti-PMSG to the authors when this evaluation is over.
→Since we did not have an eCG antibody for Western blotting, we inserted a myc-tag between the first and second amino acids and detected the recombinant protein using an anti-myc tag antibody. Additionally, we confirmed that the addition of the myc-tag did not affect the functionality, as the recombinant protein reacted normally in the PMSG ELISA Kit sold by R&D Systems.
→However, since there was no way to purchase this antibody, we added a myc-tag instead. If it is possible to provide the anti-PMSG antibody as you suggested, we would be very grateful, and we believe it would be highly beneficial to our research.

Round 3
Reviewer 1 Report
Comments and Suggestions for Authors
The authors have adequately answered most of the issues I raised.
The lack of O-glycosylation at the beta-CTP level will undoubtedly be a problem for the in vivo use of recombinant eCG produced in CHO GG44 cells for replacing natural eCG (PMSG).
Nevertheless, the data in this manuscript are of interest and must become accessible to the scientific community in the field.